# Theoretical and Experimental Investigations of Diffraction Characteristics Influenced by Holographic Reciprocity Effect in PQ/PMMA Polymers

**DOI:** 10.3390/polym15061486

**Published:** 2023-03-16

**Authors:** Peng Liu, Xiudong Sun

**Affiliations:** 1College of Physics and Electronic Engineering, Sichuan Normal University, Chengdu 610101, China; 2Institute of Modern Optics, School of Physics, Key Laboratory of Micro-Nano Optoelectronic Information System, Ministry of Industry and Information Technology, Key Laboratory of Micro-Optics and Photonic Technology of Heilongjiang Province, Harbin Institute of Technology, Harbin 150001, China; 3Collaborative Innovation Center of Extreme Optics, Shanxi University, Taiyuan 030006, China

**Keywords:** holographic reciprocity effect, holographic reciprocity matching, volume holographic storage

## Abstract

We propose the holographic reciprocity effect (HRE) to describe the relationship between the exposure duration (ED) and the growth rate of diffraction efficiency (GRoDE) in volume holographic storage. The HRE process is investigated experimentally and theoretically in order to avoid the diffraction attenuation. Herein, introducing the medium absorption, we present a comprehensive probabilistic model to describe the HRE. PQ/PMMA polymers are fabricated and investigated to reveal the influence of HRE on the diffraction characteristics through two recording approaches: pulsed exposure with nanosecond (ns) level and continuous wave (CW) exposure at the millisecond (ms) level. We obtain the holographic reciprocity matching (HRM) range of ED in PQ/PMMA polymers with 10^−6^~10^2^ s level and improve the response time to microsecond (μs) order with no diffraction deficiency. This work can promote the application of volume holographic storage in high-speed transient information accessing technology.

## 1. Introduction

With the arrival of the big data era, the demand for information storage density and access rate is gradually increased [1,2,3]. Since information diversity is an important resource for internet interaction and artificial intelligence, the development of low-cost, easily fabricated and compact information storage technology is urgently required, which will drive our lives into the next information age [4,5]. Mass storage is the core technology in the era of digital economy. The traditional two-dimensional data storage modes, such as magnetic storage, optical storage and semiconductor storage, have been widely researched. Therefore, the improvement of storage capacity is limited, and cannot meet the main needs of today’s social development for data storage. Volume holographic storage is a kind of three-dimensional storage technology which depends on the physical and chemical properties of photorefractive materials to access information [6,7]. Unlike the traditional storage methods with unit systems, it can process information in pages, which exhibits the advantages of high storage density, fast data transmission and fast addressing rate. The biggest feature of this technology is the excellent storage density, which can solve the technical bottleneck of improving the storage density, and it has great competitiveness in the application of information storage [8,9,10]. The selection of recording medium is one of the main factors affecting the holographic performance. Looking for new high-performance storage materials is an effective way to improve the holographic storage properties. Among different holographic storage media, PQ/PMMA polymers [11] exhibit unique holographic properties with high storage density, a simple fabrication method, negligible shrinkage and controllable size [12,13,14]. However, the response time of this material, which is approximately tens of seconds, has not been effectively improved. In order to complement this deficiency, we adopt high-energy pulsed exposure during the holographic recording. When the exposure duration is shortened, the holographic reciprocity failure (HRF) phenomenon will occur. To balance the response time and diffraction efficiency, we are looking to find the mismatch threshold of holographic reciprocity effect (HRE) while shortening the exposure duration. With the rapid development of various industries, there is also a new demand for an improved data access rate. Therefore, improving the response time of PQ/PMMA polymers plays a key role in their future practical progress.

Using high-energy short-pulsed exposure is an effective way to shorten the response time in holographic storage [15]. In our previous work, we successfully recorded holographic gratings in PQ/PMMA polymers under ns pulsed exposure using a dark diffusion enhancement process [15,16]. Compared with continuous wave (CW) exposure, the diffracted intensity (DI) of PQ/PMMA was declined by pulsed exposure. The main reason for this discrepancy was that the HRE occurred while shortening the ED too much [17]. The HRE can be defined as the relationship between the exposure duration (ED) and the growth rate of diffraction efficiency (GRoDE). This effect can be divided into two parts: holographic reciprocity matching and failure. If the GRoDE remains constant with the variations in ED, the holographic reciprocity matching (HRM) occurs, while if the GRoDE decays with the ED, the HRF emerges. This phenomenon has been observed in many holographic recording mediums, such as silver halide [18], photorefractive polymers [19] and photopolymers [20]. The limiting conditions of HRE need to be further studied, which will contribute to the development of rapid volume holographic storage and optical applications of photopolymers.

Our research aims to investigate the influence of HRE on holographic performances. In the experiment, the diffraction characteristics of PQ/PMMA polymers are examined with different magnitudes of ED. Two exposure approaches (pulsed and CW illumination) are adopted to generate one-time exposure under the ns and ms level. We mainly investigate the temporal intensity of holographic gratings after one-shot pulsed exposure. A 6 ns pulsed laser and a CW laser, both at 532 nm wavelength, are adopted to generate single pulse exposure under ns and ms level, respectively. PQ/PMMA polymers fabricated via a thermal polymerization method with a thickness of 1–3 mm are used to examine the HRM properties by comparing the proportional relationship between the ED and the DI. In theory, a corresponding probabilistic model of HRE with medium absorption is proposed by coupling the exposure density with the absorption coefficient. This theoretical model can effectively estimate the range of HRM on PQ/PMMA polymer materials. This work presents a foundation for improving the reconstruction quality of rapid holographic storage in future applications.

## 2. Materials and Methods

We adopt the traditional thermal polymerization method to fabricate our samples, PQ/PMMA polymers [21,22]. This polymeric compound contains three components: the host matrix Poly (methyl-methacrylate) (PMMA, Tianjin chemicals, Tianjin, China), the photo-initiator Phenanthrenequinone (PQ, Sigma-Aldrich, St. Louis, MI, USA) and the thermos initiator Azo-di-iso-butyro-nitrile (AIBN, Tianjin chemicals, China). The whole preparation strategy consists of three steps: Pre-polymerization: the mixture of PQ (0.1 wt%) and AIBN (0.05 wt%) dissolved in Methyl-methacrylate (MMA) solvent is pre-polymerized at 60 °C for 3 h. In this step, the solvent and components are fully mixed to produce a small amount of free radicals, and the nitrogen generated by the thermal decomposition of AIBN is slowly removed. High-temperature polymerization: After filtering, the solution is initiated in the incubator at 85 °C for 15 min. In this step, a large number of free radicals are generated in the solution by raising the ambient temperature, and a drastic polymerization reaction is carried out. Low-temperature baking: the viscous liquid maintains at 60 °C for 72 h for solidification. In the last step, the main chemical reaction is the chain polymerization of the monomers, which can increase the hardness of our samples. After that, samples with the diameter of 6 cm and the thickness of 1–3 mm are prepared by grinding and polishing. 

In the experiment, a twined holographic storage system is set up, as shown in Figure 1. In this system, a 532 nm CW source is firstly coupled into the optical path of pulsed holographic recording. The coupled beams can be interfered inside the medium by the splitting of PBS. Electronic controlling shutters (ECS) with the accuracy of 20 ms are adopted to adjust the exposure flux of the CW source. Meanwhile, a pulsed laser with 532 nm wavelength and 6 ns duration is applied to rapid holographic recording. The whole system can be divided into two parts: recording and reconstruction. In the recording process, two exposure approaches are adopted, i.e., the CW and pulsed exposure. In the CW exposure process, we keep S3 on and S2 off, while S1 is controlled by ECS to generate one-time exposure with different ms level duration, as shown in Figure 1a. The ms level pulsed exposure is divided into two beams with equal intensity, the same polarization and constant phase difference. These two beams begin to meet on the surface of recording material and generate interference in PQ/PMMA polymers to record the volume holographic grating. In the pulsed exposure process, we keep S2 and S3 on and S1 off to generate the holographic diffracted gratings inside the material with ns pulsed exposure, as shown in Figure 1b. After recording, a reconstruction system with CW exposure can be applied to examine the diffraction characteristics of two recording approaches, where we keep S1 on and S2 and S3 off, as shown in Figure 1c. The holographic diffracted grating recorded in PQ/PMMA materials is irradiated by the same optical path as the recording beam to generate the diffraction effect, and the holographic performance is evaluated by the intensity of diffracted light. Compared with pulsed exposure, the CW light can provide stable exposure energy, which indicates the readout intensity of diffracted gratings can be more constant. Based on this system, the HRE on PQ/PMMA polymers under different duration levels can be investigated experimentally. It is worth noting that the reason for choosing 532 nm laser as the recording and reconstruction light source is that the PQ/PMMA material has good absorption in this wavelength, and will not produce a holographic scattering effect [23,24] due to excessive absorption, which may affect its holographic performance.

In the theoretical analysis of the HRE, we consider the influence of material thickness. A variation in thickness will lead to differences in the absorption coefficient. Hence, we begin by measuring the absorptive properties of PQ/PMMA polymers with different thicknesses, as shown in Figure 2. The absorption spectrum of our samples is measured in the visible light range by a UV-3100 spectro-photometer. Figure 2a describes the initial absorption spectrum of 1–3 mm samples at the range of 500~600 nm. The specific absorption coefficient at 532 nm in 1–3 mm PQ/PMMA samples is 0.175, 0.213 and 0.313, respectively. This characterization result indicates that the light absorption intensity of PQ/PMMA materials is influenced by the sample thickness. The light absorption efficiency will be enhanced with the increment of material thickness. Figure 2b depicts temporal evolution of absorption coefficient during exposure in PQ/PMMA polymers. The exposure power in the experiment is set to 15 mW/cm^2^, which is as same as the exposure intensity of the holographic recording process with the ms level. The absorption coefficient measured in the experiment is defined as α=lgIiIt, where $I_{i}$ and $I_{t}$ represent the incident intensity and transmission intensity, respectively [25]. The experimental results of the absorbance trend in 1–3 mm PQ/PMMA polymers are depicted by the scattering plots in Figure 2b. To quantify the absorption coefficient, an exponential fitting formula $y=y_{0}+A_{1}\exp\left[-\left(x-x_{0}\right)/\tau \right]$ generated by the software OriginLab 2021 is applied to match the scattering points, which is reflected by black double-solid lines in Figure 2b. It can be found that the absorption coefficient discrepancy before and after exposure is also influenced by the sample thickness, which indicates that the material thickness affects the degree of diffused and polymerized reactions inside the material. Here, Equations (1)–(3) describe the attenuation trend of absorption coefficient in different thicknesses, respectively. The parameter of $\alpha_{1}$, $\alpha_{2}$ and $\alpha_{3}$ represents the absorption coefficient in 1–3 mm thick samples, while *t* stands for the exposure time. The fitting equations will be applied to the optimization of our theoretical model.
(1)$$\alpha_{1}\left(t\right)=0.066+0.11\times e^{\left(-\left(t-2.55\right)/38.18\right)}$$
(2)$$\alpha_{2}\left(t\right)=0.084+0.13\times e^{\left(-\left(t-4.09\right)/56.37\right)}$$
(3)$$\alpha_{3}\left(t\right)=0.122+0.19\times e^{\left(-\left(t-4.82\right)/73.18\right)}$$

## 3. Results and Discussion

### 3.1. Theoretical Probabilistic Model on the HRE

We propose a probabilistic model to assess the HRE, where the formation mechanism of diffracted grating [26,27] in PQ/PMMA polymers needs to be discussed. Two main reactions affect the grating formation process in PQ/PMMA polymers: photoinitiation and polymerization, as shown in Equations (4) and (5). This equation depicts the concentration variations of each component during photopolymerization process. The parameter $k_{T}$, $k_{R}$ and $k_{P}$ represents the duration of initiation, reduction relaxation and polymerization, respectively. The photoinitiation process in holographic recording is described in Equation (4), while the polymerization process is expressed in Equation (5).
(4)$$\mathrm{PQ}+hv\overset{k_{T}}{\underset{k_{R}}{⇄}}{\left[\mathrm{PQ}\right]}^{*}+\mathrm{PMMA}/\mathrm{MMA}$$
(5)$${\left[\mathrm{PQ}\right]}^{*}+\mathrm{PMMA}/\mathrm{MMA}\overset{k_{P}}{\to }\mathrm{PQ}-n\mathrm{MMA}$$

In the first step, the transition relaxation and reduction relaxation of photosensitizers are defined as random variables *F* and *R*. Meanwhile, in the second step, the polymerization relaxation is also a random variable *P*. Their exponential equations can be described as:(6)$$F(t)=Ch\{k_{T}\leq t\}=1-e^{-Dt}$$
(7)$$P(t)=Ch\{k_{P}\leq t\}=1-e^{-\alpha t}$$
(8)$$R(t)=Ch\{k_{R}\leq t\}=1-e^{-\beta t}$$
where *D* reflects exposure density inside the material, which is defined as the reciprocity coefficient. This can be regarded as the main parameter to judge the influence of HRE on diffraction characteristics. The corresponding probability density functions of *F*(*t*), *P*(*t*) and *R*(*t*) are *f*(*t*), *p*(*t*) and *r*(*t*). Additionally, $\alpha^{-1}$ and $\beta^{-1}$ represent the average of *P*(*t*) and *R*(*t*). The Ch{} is the probability of events. We set event *H* as the occurrence of the whole photopolymerization, which can be divided into two parts: event *A*, in which photosensitizers directly polymerize after initiation. Additionally, in event *B*, photosensitizers relax to the ground state after initiation and polymerize after the secondary excitation. The probability distribution densities of *A* and *B* are *u*(*t*) and *M*(*t*), which can be described as
(9)$$u(t)=f(t)*p(t)=\frac{-D\alpha (e^{-Dt}-e^{-\alpha t})}{D-\alpha }=\frac{-D\alpha (e^{-E}-e^{-\alpha t})}{D-\alpha }$$
(10)$$M(t)=Ch\{B\leq t\}=\frac{\left(1-e^{-\frac{E(\beta^{-1}-\alpha^{-1})}{t}}\right)\left(1-e^{-E}\right)}{e^{-\frac{E(\beta^{-1}-\alpha^{-1})}{t}}+{\left(1-e^{-\frac{E(\beta^{-1}-\alpha^{-1})}{t}}\right)}^{2}}$$

Hence, the probability of event *H* (*H* = *A* + *B*) can be given, as shown in Equation (11), where the parameter *E* is the exposure energy of the incident beam, and the symbol * means the conjugate operation.
(11)D(t,E)=Ch{H≤t}=u(t)∗M(t)

According to the reciprocity coefficient D(t,E) derived from Equation (11), it is implied that the HRE is related to the exposure time *t* and energy *E*. However, the photosensitizers in the exposure area will be consumed with the increase in the exposure time, which means the absorption coefficient will be attenuated in the holographic recording process. In order to avoid the influence of this phenomenon, we introduce the evolution equation of the absorption coefficient caused by the medium absorption, as shown in Equations (1)–(3). Due to the complexity of the multistage chain reactions in PQ/PMMA polymers, we obtain the absorption coefficient equations through exponential fitting of our experimental results. Then, the normalized absorption variations are coupled with the reciprocity coefficient D(t,E) via the parameter *t*. Hence, the final reciprocity coefficient is derived, as shown in Equation (12), where $\alpha_{n}\left(t\right)$ represents the evolution equation of the absorption coefficient and $\alpha_{n}\left(0\right)$ stands for the initial value of the absorption coefficient before holographic recording.
(12)$$D_{\alpha }(t,E)=\left[\alpha_{n}\left(t\right)/\alpha_{n}\left(0\right)\right]\times D(t,E){}_{}\;n=1,2,3\ldots$$

In our simulation, the polymerization and reduction relaxation are set to the magnitude of 10^−6^ and 10^−2^ according to the fitting results in refs. [28,29], respectively. We simulate and analyze the influence of exposure energy and material thickness on HRE. Figure 3a depicts the nonlinear relationship between the exposure energy and the reciprocity coefficient. The total temporal range is set to 10^−16^~10^2^ s. The GRoDE will stay constant if the reciprocity coefficient remains stable, which means HRM occurs; otherwise, the HRF will emerge. It is found that the range of HRM in PQ/PMMA polymers is approximately 10^−6^~10^2^, as shown in Figure 3. Meanwhile, the matching range can be extended by raising the exposure energy, which indicates higher exposure energy can reduce the influence of HRF on rapid holographic storage. Figure 3b illustrates the influence of material thickness on the HRE. The material thickness determines the absorption coefficient in recording medium, which implies it can affect the authentic absorptive rate of the exposure. Although the influence of material thickness is not as severe as the exposure energy, it can still be regarded as an optimization indicator to improve the range of HRM. Hence, the effect of HRF on holographic properties can be effectively avoided by controlling the material thickness and the exposure energy in holographic recording. However, if we want to broaden the range of HRM, a kind of new material with a faster polymerization rate needs to be found, which will be our main research direction in the future.

### 3.2. Experimental Results on the HRM and the HRF

In order to further investigate the HRE in PQ/PMMA polymers, we experimentally investigate the diffraction characteristics of our materials after one-time exposure [30], which means the intensity of each diffraction grating is measured once and the only variable is the duration of each exposure. The HRE is evaluated by the GRoDE, where the diffraction efficiency can be defined as the ratio of exposure intensity of the diffracted and reconstructed reference beam [31] and the GRoDE is the slope of diffraction efficiency at each recording point. Additionally, the response time is achieved by exponential fittings of temporal diffraction curves. Two temporal ranges are examined in our experiment, i.e., 10^−7^~10^−5^ and 10^−2^~10^2^, by using a ns pulsed laser and a ECS combined CW laser system, respectively. Notably, our measurements are described via two coordinate layouts (linear and logarithmic abscissa) for more convenient comparison with our theoretical results.

Figure 4 depicts the temporal diffraction characteristics with the duration of 10^−7^~10^−5^ s. In the experiment, the ns pulsed exposure laser is used to record the holographic grating, while the CW laser is adopted as the reconstructed source. The pulsed exposure energy is set to 2 mJ, while the reconstruction energy is 3 mW/cm^2^. In our previous work, we demonstrated that the dark diffusion enhancement is very slight after ns pulsed exposure [13]. Hence, we can obtain different EDs by adjusting exposure times at the maximum frequency of ns pulsed laser (10 Hz) without the DI floating. In this way, the ED in the experiment is set to 6, 12, 30, 48, 60, 120, 300, 600 and 1200 × n, (n = 1, 2, 3… 15) ns. The real-time diffraction efficiency after one-time exposure is measured and analyzed with different material thicknesses (1–3 mm thick PQ/PMMA polymers), as shown in Figure 4a,b. According to the experimental results, the variation trend of DI can be divided into three stages. At the beginning, the DI is almost zero, since the one-time exposure is too swift to excite the photosensitizers. In the second stage, the DI will increase linearly with the ED, where the HRM appears. At last, the phenomenon of photo-bleaching will occur when the organic dye photosensitizers are consumed in the exposure area. Therefore, the DI will slightly decay when the exposure flux is excessive, which is mainly due to the deficiency of photosensitizers content. The first and the last stage indicates the HRF occurs. In order to show the range of HRM more intuitively, the diffraction slope gradient is fitted and depicted, as shown in Figure 4c,d. The height difference between the two plains represents the loss degree of HRF. It is indicated that raising the medium effective thickness can improve the matching stability of HRE, where the effective thickness can be defined as the material thickness that contributes to the formation of holographic grating. This is affected by the polymerization degree and the photosensitizer concentration inside the material. Commonly, the effective thickness is less than the actual thickness. The range of HRM in PQ/PMMA polymers is approximately 10^−6^~10^−5^ s under ns pulsed exposure with the duration of 10^−7^~10^−5^ s, which indicates the response time of PQ/PMMA polymers can be developed to μs level with no influence of the HRE. This result also exhibits the recording ability of transient information in PQ/PMMA materials by increasing the exposure energy of each pulse, which also broadens the response range of volume holographic storage technology.

Figure 5 exhibits the temporal diffraction characteristics with the duration of 10^−2^~10^2^ s. The one-time exposure of ms magnitude is achieved via the coordination of the 532 nm CW laser and the ECS. In the experiment, the S2 is closed to prevent ns pulsed exposure, while the S1 controls the ED and the S3 can switch the mode between the recording and the reconstruction process. The ED in the experiment is set to 20, 50, 100, 200, 400 × n, (n = 1, 2, 3… 100) ms. The DI of PQ/PMMA polymers under ms level exposure is displayed in Figure 5a,b. It is indicated the range of HRM can be extended extensively by increasing the ED to ms level, which can prolong it from 10^−6^~10^−5^ s to 10^−2^~10^2^ s. Notably, when the ED is too long (>10^2^ ms), the photosensitizers in the exposure area are completely consumed, while the photosensitizers in the unexposed area cannot diffuse to the exposure area immediately. This will also cause the occurrence of HRF. Meanwhile, the loss degree of HRF is analyzed by the diffraction slope gradient, as shown in Figure 5c,d. Compared with the height discrepancy between peak and trough in Figure 4d and Figure 5d, the floating degree of the diffraction slope gradient decreases from 10^6^ to 10^−3^, which implies the holographic grating stability can be improved by enlarging the exposure flux. Although holographic gratings can be built in PQ/PMMA polymers by increasing the exposure energy, shorter exposure durations will also affect the stability of grating formation. How to maintain the stability of grating formation after short exposure needs to be further studied.

To verify the probabilistic model, we compare the experimental results with our theoretical simulations, as shown in Figure 6. It is indicated that the range of HRM fits well with the experimental results, where the HRM of PQ/PMMA polymers is approximate 10^−6^~10^2^ s. However, the range of HRF does not agree well with our experimental results. This discrepancy may be caused by two aspects. One is the deviation of material thickness between the experiment and the theoretical simulation, which further affects the absorption coefficient. The other is the fluctuation of various reaction rates during photopolymerization. Hence, the probabilistic model can estimate the mismatch threshold of HRE in different ratios of photosensitizer components and can also be regarded as the theoretical basis for investigating the diffraction characteristics in rapid volume holographic storage.

We apply the pulsed and CW exposure method to examine the imaging quality of reconstructed holograms, as shown in Figure 7. Two main indicators are adopted to access the imaging quality, i.e., the grating strength and the signal-to-noise ratio (SNR). The grating strength can be calculated from the square root of diffraction efficiency, which can evaluate the fringe contrast of the reconstructed grating. The expression of SNR is $\mathrm{SNR}=10\cdot \mathrm{lg}\left(P_{S}/P_{N}\right)$, where $P_{S}$ and $P_{N}$ represent the energy of signal and noise. The SNR can reflect the absence of information carried by the reconstructed grating. In the experiment, the 1 mm thick PQ/PMMA sample is used to record holograms by multiple exposures. The maximum grating strength of holograms by pulsed and CW exposure recording is 0.45 and 0.52, while the SNR is 54.55 and 60.13. The experimental results indicates that the HRE occurs under ns pulsed exposure; both the grating strength and SNR lead to attenuation. Hence, the reconstruction quality of information and data can be improved by estimating the mismatch threshold and avoiding HRF during rapid holographic storage.

## 4. Conclusions

In this paper, the holographic reciprocity effect has been comprehensively researched in theory and experiments. A probabilistic model of exposure density is coupled with the absorption equations of different material thickness, which can reflect the influence of exposure duration and material thickness on the holographic reciprocity effect. The holographic reciprocity failure can be effectively avoided by increasing the exposure energy and adjusting the material thickness. A twined holographic storage system is applied to examine the diffraction characteristics in PQ/PMMA polymers after one-time exposure under different duration levels (ms and μs). The range of holographic reciprocity matching is approximately 10^−6^~10^2^ s, which also indicates the response time can be shortened to μs magnitude in PQ/PMMA polymers with no diffraction deficiency. Meanwhile, the matching range of holographic reciprocity can be roughly estimated by our probabilistic model. Through our research, the holographic reciprocity effect is pointed out to be the main difficulty in transient holographic storage. By finding an effective way to solve this problem, we can effectively avoid the impact of holographic reciprocity failure and promote the practical development of volume holographic storage in rapid information access technology.

## Figures and Tables

**Figure 1 polymers-15-01486-f001:**
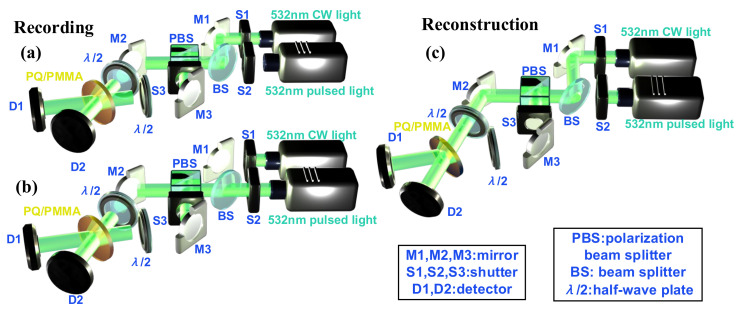
Holographic storage system. (**a**) The recording process under ms level, (**b**) the recording process under ns level, (**c**) the reconstruction process by CW exposure.

**Figure 2 polymers-15-01486-f002:**
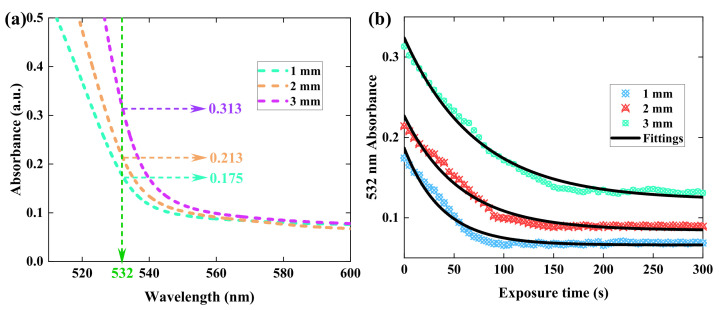
(**a**) The absorption spectrum of PQ/PMMA polymers with 1–3 mm thickness, (**b**) experimental results of absorbance variations in 1–3 mm thick PQ/PMMA polymers.

**Figure 3 polymers-15-01486-f003:**
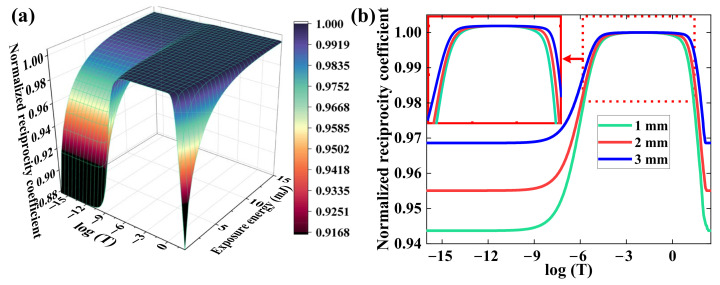
The evolution of normalized reciprocity coefficient with (**a**) different exposure energy (0–15 mJ), and (**b**) different material thickness (1–3 mm).

**Figure 4 polymers-15-01486-f004:**
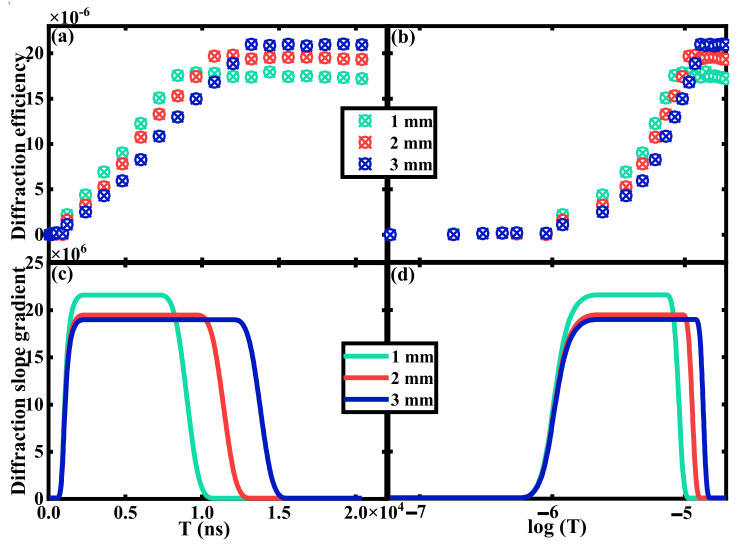
Temporal diffraction characteristics of HRE by ns level exposure with (**a**) linear abscissa (T), (**b**) logarithmic abscissa (log(T)), and the diffraction slope gradient with (**c**) linear abscissa (T) and (**d**) logarithmic abscissa (log(T)).

**Figure 5 polymers-15-01486-f005:**
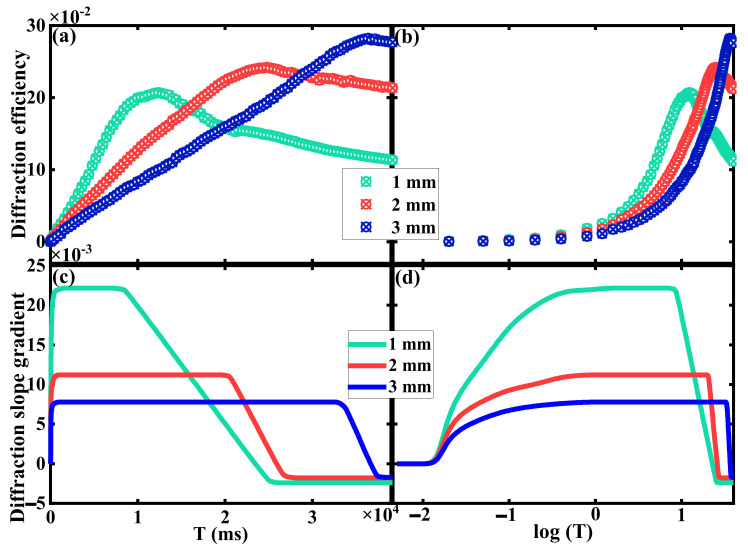
Temporal diffraction characteristics of HRE by ms level exposure with (**a**) linear abscissa (T), (**b**) logarithmic abscissa (log(T)), and the diffraction slope gradient with (**c**) linear abscissa (T) and (**d**) logarithmic abscissa (log(T)).

**Figure 6 polymers-15-01486-f006:**
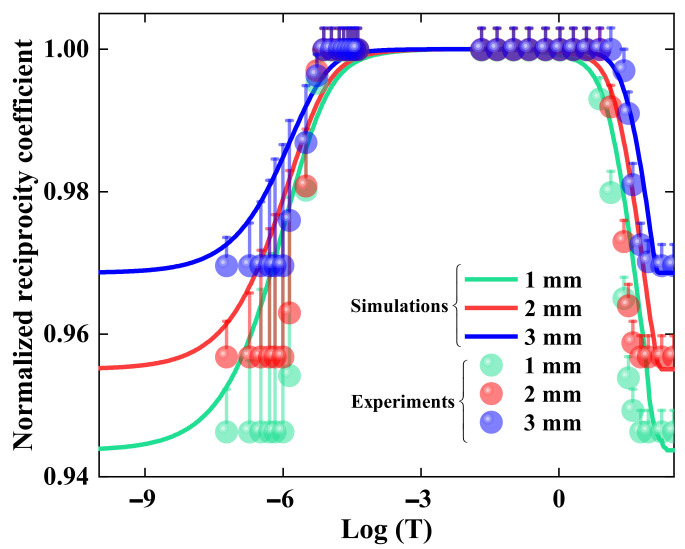
Comparisons of theoretical and experimental results on HRE in PQ/PMMA polymers.

**Figure 7 polymers-15-01486-f007:**
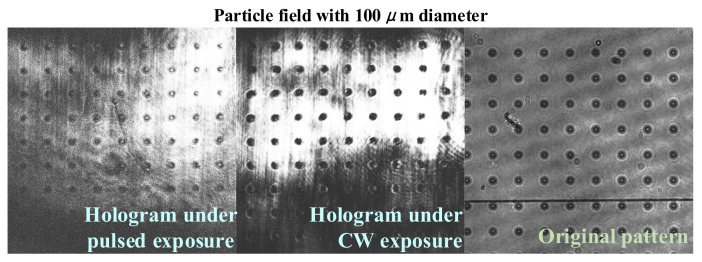
Reconstructed holograms of 100 μm particle field with pulsed and CW exposure method.

## Data Availability

The data presented in this study are available on request from the corresponding authors.

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
