# Peer review of "Theoretical and Experimental Investigations of Diffraction Characteristics Influenced by Holographic Reciprocity Effect in PQ/PMMA Polymers"

_polymers, 2023, doi:10.3390/polym15061486_

Round 1
Reviewer 1 Report
The research is aimed to investigate the influence of holographic reciprocity effect (HRE) on holographic performances. The diffraction characteristics of PQ/PMMA polymers are examined with different magnitudes. Two exposure approaches (pulsed and CW illumination) are adopted to generate one-time exposure under ns and ms level. A corresponding probabilistic model of HRE with medium absorption is proposed by coupling the exposure density with absorption coefficient. This work presents a foundation on improving the reconstruction quality of rapid holographic storage in future applications.
The paper can be published in the present form.
Author Response
Thanks for the reviews and we enrich the content of the article and make it more readable, but the overall content and framework has not been influenced.
Reviewer 2 Report
Presented manuscript reports further results of on-going authors' research of holographic data storage. In general manuscript is well-organized and written and presents interesting results. Nevertheless, to improve its readability I kindly ask authors to address the following points:
1. Please give the explanation for every abbreviation in main text (not only in abstract) where it appears for the first time, e.g. HRF, HRE, DI (line 50), ED, MMA (line 74) etc.
2. Lines 125-139 contain the part of template text. Please, remove and double check that no parts of ‘Materials and Methods’ section were missed.
3. Lines 146-150. I guess, the colored brakets will not be distinguishable in printed version of manuscript. Please, check if bold, italic of under/upper-lined fonts are appropriate for highlighting.
Author Response
- Please give the explanation for every abbreviation in main text (not only in abstract) where it appears for the first time, e.g. HRF, HRE, DI (line 50), ED, MMA (line 74) etc.
Answer: We give the detail explanation for the abbreviation in the content when it firstly appears.
- Lines 125-139 contain the part of template text. Please, remove and double check that no parts of ‘Materials and Methods’ section were missed.
Answer: We checked and deleted the part of template on ‘Materials and Methods’ section in our manuscript.
- Lines 146-150. I guess, the colored brakets will not be distinguishable in printed version of manuscript. Please, check if bold, italic of under/upper-lined fonts are appropriate for highlighting.
Answer: To make this equation easier to understand, we split this formula into two parts, as shown in the word file.
